# Production of Circularly Permuted Caspase-2 for Affinity Fusion-Tag Removal: Cloning, Expression in *Escherichia coli*, Purification, and Characterization

**DOI:** 10.3390/biom10121592

**Published:** 2020-11-24

**Authors:** Monika Cserjan-Puschmann, Nico Lingg, Petra Engele, Christina Kröß, Julian Loibl, Andreas Fischer, Florian Bacher, Anna-Carina Frank, Christoph Öhlknecht, Cécile Brocard, Chris Oostenbrink, Matthias Berkemeyer, Rainer Schneider, Gerald Striedner, Alois Jungbauer

**Affiliations:** 1ACIB-Austrian Centre of Industrial Biotechnology, Muthgasse 18, 1190 Vienna, Austria; monika.cserjan@boku.ac.at (M.C.-P.); petra.engele@uibk.ac.at (P.E.); Christina.Kroess@uibk.ac.at (C.K.); julian.loibl@boku.ac.at (J.L.); andreas.fischer@boku.ac.at (A.F.); florianbacher@gmx.at (F.B.); anna.frank@boku.ac.at (A.-C.F.); christoph.oehlknecht@boku.ac.at (C.Ö.); chris.oostenbrink@boku.ac.at (C.O.); Rainer.Schneider@uibk.ac.at (R.S.); gerald.striedner@boku.ac.at (G.S.); 2Department of Biotechnology, University of Natural Resources and Life Sciences, Vienna, Muthgasse 18, 1190 Vienna, Austria; 3Institute of Biochemistry, Center for Molecular Biosciences Innsbruck (CMBI), University of Innsbruck, 6020 Innsbruck, Austria; 4Institute of Molecular Modeling and Simulation, University of Natural Resources and Life Sciences, Vienna, Muthgasse 18, 1190 Vienna, Austria; 5Biopharma Process Science Austria, Boehringer Ingelheim RCV GmbH & Co KG, Dr. Boehringer-Gasse 5-11, 1121 Vienna, Austria; cecile.brocard@boehringer-ingelheim.com (C.B.); matthias.berkemeyer@boehringer-ingelheim.com (M.B.)

**Keywords:** affinity tag removal, fusion protein, proteases, recombinant protein, solubility enhancing tag, His-tag, platform process, circular permutation

## Abstract

Caspase-2 is the most specific protease of all caspases and therefore highly suitable as tag removal enzyme creating an authentic N-terminus of overexpressed tagged proteins of interest. The wild type human caspase-2 is a dimer of heterodimers generated by autocatalytic processing which is required for its enzymatic activity. We designed a circularly permuted caspase-2 (cpCasp2) to overcome the drawback of complex recombinant expression, purification and activation, cpCasp2 was constitutively active and expressed as a single chain protein. A 22 amino acid solubility tag and an optimized fermentation strategy realized with a model-based control algorithm further improved expression in *Escherichia coli* and 5.3 g/L of cpCasp2 in soluble form were obtained. The generated protease cleaved peptide and protein substrates, regardless of N-terminal amino acid with high activity and specificity. Edman degradation confirmed the correct N-terminal amino acid after tag removal, using Ubiquitin-conjugating enzyme E2 L3 as model substrate. Moreover, the generated enzyme is highly stable at −20 °C for one year and can undergo 25 freeze/thaw cycles without loss of enzyme activity. The generated cpCasp2 possesses all biophysical and biochemical properties required for efficient and economic tag removal and is ready for a platform fusion protein process.

## 1. Introduction

A perfect protease for fusion tag removal has not yet been found. Such enzymes should be highly specific, soluble, and stable and should be easy to manufacture to reduce the economic burden. Therefore, they should be at best expressible in *E. coli* in soluble form at high concentration. This would also simplify the purification of such a protease. In this work, we describe how to arrive close to such a perfect protease by circularly permuting human caspase-2.

The production of recombinant proteins is still cost and time-consuming because the diverse characteristics of proteins require complex and tailor-made processes with a high number of unit operations in up- and downstream processing. A fusion technology would offer a platform solution for the production of these proteins. Platform technologies could accelerate the development efforts and time and reduce costs for laboratory and industrial production of proteins [1]. Such technologies should then ideally combine efficient soluble expression, an affinity tag enabling fast and simple purification and a system for efficient tag removal.

Several strategies for tag removal are available, but large-scale recombinant protein production with fusion protein technology is not yet standard in industry. For therapeutic applications, the removal of tags is highly recommended in order to avoid immune-response against the tag. An aberrant or non-natural N-terminus of the protein of interest (POI) requires additional preclinical data, when such a protein is used for therapeutic applications. A non-native N-terminus can be produced through extraneous amino acids after tag removal or through cryptic cleavage sites [2,3,4]. It has also been observed that a tag or non-native N-terminus may alter the conformation of the POI [5,6,7,8]. Therefore, it is necessary to create an authentic N-terminus when a protein is studied for structure function properties. The most commonly used proteases for tag removal are endopeptidases such as factor Xa, thrombin, tobacco etch virus protease, SUMO protease, and enterokinase [1,9]. However, all of these proteases are either expensive, inefficient, or show unspecific cleavage [10,11]. A cleavage site that leaves residual amino acids at the N-terminus of the product or the need for special buffer conditions are not favorable for all POIs [12].

In this work, caspase-2 was selected as it is unique among caspases in recognizing a specific pentapeptide instead of a tetrapeptide with high affinity [13]. Cleavage occurs immediately after the P1 amino acid of the recognition sequence [13]. This increased length of the recognition site results in significantly increased specificity. VDVAD is considered the preferred cleavage site of caspase-2 [14], corresponding to the P1–P5 sites using the Schechter and Berger protease nomenclature [15]. In caspases, the prime side of the substrate (i.e., the amino acids directly C-terminal to the cleavage site) can influence catalytic activity by steric hindrance, especially the P1′ residue [16,17]. Caspase-2 is synthesized as a relatively inactive single-chain zymogen and is in vivo activated into a dimer of a small and a large subunit heterodimer [18]. Active caspases form dimers of heterodimers. To obtain active recombinant caspases, the subunits can be expressed separately and mixed after purification [19], or the procaspase is expressed and activates itself autocatalytically [20]. Furthermore, as caspases are active in *E. coli*, they could also cleave bacterial proteins and negatively influence growth and yield [21]; thus high specificity, as in the case of caspase-2, is a clear advantage for its efficient production in *E. coli*.

Three systems using caspases [22,23,24] have been developed. Even though these caspase-based tag cleaving systems have been published more than ten years ago, they have not been adopted into the common repertoire of protein purifications. In addition to the difficult production of caspases, the low substrate specificities of caspase-3 and caspase-6 [21] might have prevented widespread use.

Tagging of proteins is generally done for equipping a protein with additional functions [25,26]. A tag can be used as generic means of purification, as generic means of detection or to increase soluble expression. Polyhistidine tags (His-tag) are the most commonly used purification tags [27,28,29,30,31], due to their small size and the high affinity to readily available metal chelate chromatography materials, which even allows purification under denaturing conditions. Many tags have been suggested for enhancing soluble expression of proteins [32]. Zhang et al. extracted a series of peptide sequences from the T7 bacteriophage, which efficiently propagates in *E. coli* cells. Several of these peptides showed remarkable improvement of soluble protein expression [33].

In this study, we describe (i) the generation of a circularly permuted caspase-2, (ii) the effect of a solubility tag on expression level, (iii) an optimized model-based controlled fermentation process, (iv) a swift and efficient purification process of the generated enzyme, and (v) the enzyme characteristics with respect to kinetics and stability. A constitutively active cpCasp2 could be an ideal protease for biotechnological applications. Such a variant can be generated through circular permutation (cp), which is the covalent linkage of native N- and C-terminus and the introduction of new termini elsewhere in the protein [34]. While such new variants have an altered order of amino acids, they maintain their tertiary structure. Several circularly permuted caspases have been described [34,35,36,37]. All but one described variant (uncleavable cp Caspase-3 [34]) cleave themselves at the inter-subunit linker. Therefore, the active enzyme still consists of four amino acid chains. Published examples of circularly permuted caspases focused on researching apoptosis and did not have a biotechnological use in mind, therefore aiming for a structure similar to the wild type variants.

## 2. Materials and Methods

### 2.1. Strains and Primers

For cloning purposes, chemically competent *E. coli* NEB-5α cells were purchased from New England Biolabs (NEB, Ipswich, MA, USA). Expression vectors were transformed into the *E. coli* strain BL21(DE3) (NEB). Cells were cultured and processed according to the manufacturer’s protocols. The primers were purchased from Sigma Aldrich (St. Louis, MO, USA) (Appendix A).

### 2.2. Design of Constructs

All enzymes and kits were purchased from NEB. For site specific mutations, the NEB Q5^®^ Site-Directed Mutagenesis Kit was used. For the expression systems, we cloned the sequences into the pET30a*cer* expression vector which was produced in-house [38]. The POI is under the control of a T7 promoter. See the Appendix A for the sequences of all constructs with an explanation of the subunits and tags (Appendix A).

#### 2.2.1. Wildtype Caspase-2 (wtCasp2)

The gene of human caspase-2 (UniProtKB ID P42575) without CARD (residues 170–452) and a 6H-tag C-terminal of the small subunit (Figure 1a) was designed as described elsewhere [39], codon-optimized with GeneArt online tool (Thermo Fisher Scientific, Waltham, MA, USA) and synthesized (BioCat, Heidelberg, Germany). The gene was cloned into a pET30a*cer* expression vector with *Nde*I and *Xho*I restriction sites.

#### 2.2.2. Circularly Permuted Human Caspase-2 (cpCasp2)

As shown in Figure 1b, a cpCasp2 was designed by removing the N-terminal CARD and by switching the order of large (p18, residues 170–333) and small subunit (p13, residues 334–452) of human caspase-2 (UniProtKB ID P42575). To avoid separation of the pro-peptide (residues 334–347) and the small subunit (residues 348–452), Asp^347^ was mutated to alanine. The small subunit was linked to the N-terminus of the large subunit via a glycine-serine-linker (GS). A 6H-tag was added on the N-terminus of the resulting cpCasp2, in order to facilitate purification. The protein sequence was codon optimized for *E. coli* with the GeneArt-online tool and cDNA ordered through the same platform. The gene was cloned into a pET30a*cer* expression vector with *Nde*I and *Xho*I restriction sites.

#### 2.2.3. T7AC-Tagged Circularly Permuted Caspase-2 (T7AC-cpCasp2)

A solubility tag, a peptide extension with large net negative charge, was fused to the N-terminus of the 6H-tagged circularly permuted caspase-2 resulting in T7AC-cpCasp2 (Figure 1c,d). Based on the T7A3 tag described by Zhang et al. [33], the modified T7AC tag was designed to avoid autocatalytic cleavage of the tag by cpCasp2. Glu^16^ and Glu^18^ were changed to glutamine. Full sequence LEDPERNKERKEAELQAQTAEQ. The gene that encodes the T7AC solubility tag was codon optimized for *E. coli* and synthesized by ATUM (Newark, CA, USA). The gene was cloned into the plasmid (pET30a*cer*-cpCasp2) with *XbaI* and *BsaI* restriction sites.

#### 2.2.4. Model Substrate Human Ubiquitin-Conjugating Enzyme E2 L3 (E2)

The human ubiquitin-conjugating enzyme E2 L3 (E2; UniProt ID P68036) was used as a model protein substrate. A pET30a*cer* vector (Novagen, Merck, Darmstadt, Germany) containing the E2 gene with an N-terminal caspase cleavage tag was used to express the model protein. The caspase cleavage tag consists of a hexa-histidine tag (6H), a short linker (GSG), and the caspase-2 cleavage site (VDVAD [13]). The P1′ residue is a glycine. With site directed mutagenesis, the P1′ residue was changed to all 19 canonical residues. Likewise, the cleavage site VDVAD was mutated to DEVD, EFKD, EISD, VDQQE, VDQQS, DETD-R, and DETE-R with primers listed in the Appendix A. The unprocessed protein has a size of 21.3 kDa, whereas, when the tag is cleaved off, the E2 protein itself has 19.5 kDa. This difference is big enough to visualize the cleavage products on an SDS-PAGE.

### 2.3. Bioreactor Fed-Batch Cultivations

All fermentations were performed in a 30 L (23 L net volume, 5 L batch volume) computer-controlled bioreactor (Bioengineering; Wald, Switzerland) equipped with standard control units (Siemens PS7, Siemens WinCC, Siemens AG, Munich, Germany). The pH was maintained at a set-point of 7.0 ± 0.05 by addition of 25% ammonia solution (*w/w*), and the temperature was set to 37 °C ± 0.5 °C in the batch phase and 30 °C ± 0.5 °C in the fed-batch phase. To avoid oxygen limitation, the DO level was held above 30% saturation by adjusting the stirrer speed and the aeration rate of the process air. The maximum overpressure in the head space was 1.1 bar. Foaming was suppressed by addition of 0.5 mL/L antifoam (PPG 2000) right from the beginning. The composition of the batch and the fed-batch medium used in this study is described elsewhere [40]. All chemicals were purchased from Merck (Darmstadt, Germany) in analytical grade, unless specified otherwise. All media components were added in relation to the grams of calculated CDM to be produced and for calculation the required yield coefficient Y_X/S_ of 0.3 g/g specific for BL21(DE3) was used. To prevent solubility issues, the main part of mineral compounds was already provided via the batch media as described in detail by Török et al. [41]. All cultivations were carried out without antibiotics.

For inoculation, pre-cultures were grown in synthetic media calculated to produce 3 g/L. Therefore, 1 mL of a deep-frozen working cell bank was thawed and 1 mL (OD_600_ = 1) was aseptically transferred to 400 mL medium and cultivated in 2000 mL shake flasks at 37 °C and 180 rpm until the OD_600_ reached a value of approx. 4. Then, the batch was inoculated with the pre-culture to an initial OD_600_ of 0.10 and cultivated at 37 °C. At the end of the batch phase, an exponential carbon-limited substrate feed was started. The substrate feed was controlled by increasing pump speed according to the exponential growth algorithm, X = X_0_ ∙ e^μt^, with superimposed feedback control of weight loss in the substrate tank.

#### 2.3.1. Fermentation Parameters for Production of wtCasp2 and cpCasp2 ± T7AC Solubility Tag

For these high cell density (HCD) cultivation experiments, minimal media were calculated to produce 80 g CDM in the batch medium and an additional 1373 g CDM in the feed medium, which corresponds to 4.2 doublings in the feed phase. The batch volume was fixed to 10 L and the feed to 8.6 L, resulting in 78 g/L calculated CDM. The fed-batch phase (29 h) was performed at 30 °C with an exponential feeding strategy with a consistent growth rate of µ = 0.1 h^−1^. Induction started with fed-batch phase by adding 0.5 µmol IPTG /g CDM directly to the feed-media to achieve protein production for 29 h. IPTG concentration was calculated with the theoretical final CDM.

#### 2.3.2. Fermentation Parameters for a Design of Experiment (DoE) Approach

In the DoE study, the impact of two critical process parameters (CPP) comprising the growth rate during the production phase and the induction strength were investigated. For all cultivations in the DoE approach, minimal media were calculated to produce 64 g CDM in the batch medium and for another 989 g CDM in the feed medium, which corresponds to 4 doublings in the feed phase. The batch volume was fixed to 8 L and the feed to 7.5 L resulting in 68 g/L calculated CDM. For biomass production, the first fed-batch phase was performed with an exponential feed (µ = of 0.17 h^−1^) for 1.72 generations. In a second feed-phase, different lower growth rates (0.03, 0.05 and 0.07 h^−1^) were adjusted resulting in a total feed time of 60.5 h, 39 h and 30 h, respectively. To ensure sufficient adaption to the low growth conditions, the cells grew for 0.25 generations without induction. Then, induction was performed with three different IPTG concentrations (0.5, 0.9 and 1.3 µmol/g CDM) for two generations. Nine CPP combinations in total were performed.

#### 2.3.3. Optimized Fermentation Parameters for T7AC-cpCasp2 Production

Based on the results of the DoE study, the following fermentation strategy was designed. The minimal media were calculated to produce 64 g CDM in the batch medium and for another 1537 g CDM in the feed medium, which corresponds to 4.6 doublings in the feed phase. The batch volume was fixed to 8 L and the feed to 6.8 L resulting in 108 g/L calculated CDM. For biomass production, the first fed-batch phase was performed with an exponential feed (µ = of 0.17 h^−1^) for 2.7 generations. In a second feed-phase, a lower growth rate of 0.03 h^−1^ was adjusted resulting in a total feed time of 56 h. To ensure sufficient adaption to the low growth conditions, the cells grew for 0.25 generations without induction. Then, induction was performed with a constant IPTG concentrations (0.9 µmol/g CDM) for 1.65 generations. That means an appropriate amount was added to the bioreactor at induction and again to the remaining feed medium.

### 2.4. Fermentation Monitoring

In addition to standard online monitoring (pH, stirrer speed, temperature, and pO_2_), the concentration of pO_2_ and O_2_ in the outlet air was measured with a BlueSens (Herten, Germany) gas analyzer. The first samples of the offline process parameters (OD_600_, CDM and product) were drawn from the bioreactor prior to induction. 

#### 2.4.1. Biomass Quantification

Optical density (OD_600_) of cell suspension was measured at a wavelength of 600 nm using a spectrometer model type ULTRA-SPEC (Amersham Bioscience Europe GmbH, Freiburg, Germany). Samples were diluted in PBS to ensure a measurement at a linear range from 0.1 to 0.8. Cell dry mass (CDM) was determined according to Cserjan-Puschmann et al. [42] by centrifugation of 10 mL cell suspension, resuspension in distilled water followed by centrifugation, and resuspension for transfer to a pre-weighed beaker, which was then dried at 105 °C for 24 h and re-weighed. The progress of bacterial growth was determined by calculating the total amount of biomass. The progress of bacterial growth was determined by calculating the total amount of biomass. 

#### 2.4.2. Cell Disintegration, Soluble and Insoluble Protein Fractionation, and IB Solubilization

For intracellular product analysis, fermentation samples containing approximately 1.0 mg CDM, estimated via direct measurement of the OD_600,_ were drawn every 2 h during the production phase. The calculated volume was transferred to 1.5 mL reaction tubes and centrifuged at 13,100 rcf and 4 °C for 10 min. After discarding the supernatant, the cell pellets were stored at −20 °C until analysis. Cell disintegration, fractionation of soluble and insoluble protein, and IB solubilization were performed according to Fink et al. [43]. NuPAGE Sample Reducing Agent (10×) (Invitrogen, Waltham, MA, USA) was additionally added to a concentration of 4 mM to the lysis buffer.

#### 2.4.3. SDS-PAGE Analysis

Caspase-2 variants were detected on one-dimensional sodium dodecyl sulfate polyacrylamide gel electrophoresis (SDS-PAGE) according to Stargardt et al. [44]. A ready-to-use molecular weight marker (Mark12™, Unstained Standard, Invitrogen, Waltham, MA, USA) was directly loaded as the size marker. The concentration of the soluble and insoluble fraction was calculated from densitometry analysis of SDS-PAGE gels using the software ImageQuantTL (7.0) (Cytiva, Uppsala, Sweden) via linear regression analysis. Purified T7AC-cpCasp2 (75, 50 and 25 µg/mL) was used as the standard. Two replicates of each sample were analyzed two times and the maximal coefficient of variation of 10% was used as acceptance criterion.

### 2.5. Cell Harvest, Cell Disintegration, and Clarification

The cell mass was harvested by centrifugation at 18,590 rcf for 15 min. The pellet was stored at −80 °C until further use and the supernatant was discarded. The *E. coli* harvest was solubilized using homogenization buffer: 50 mM NaPO_4_, 500 mM NaCl, 20 mM imidazole, pH 7.0. For wtCasp2, cpCasp2 and T7AC-cpCasp2, 675 g, 1011 g and 700 g cell wet mass were used, respectively. The cells were re-suspended at a concentration of 30 g cell dry mass per L. Cell lysis was performed by high pressure homogenization (Panda PLUS 2000, Gea, Düsseldorf, Germany) with two passages at 700 bar for wtCasp2 and cpCasp2. For T7AC-cpCasp2, the pressure was increased to 1400 bar. The homogenate was centrifuged (Beckman Avanti JXN-26 with JLA-10.500 rotor, Krefeld, Germany) at 18,600 rcf for 2 h. The supernatant was filtered through a 0.2 µm membrane (Kleenpak™ Capsule with Fluorodyne^®^ EX Grade EDF Membrane, Pall, New York, NY, USA).

### 2.6. Chromatographic Purification Steps

Preparative chromatography runs were performed on an Äkta Pure 25 system, equipped with a S9 sample pump (GE Healthcare, Uppsala, Sweden). The 6-His-tagged caspase-2 constructs were captured using IMAC.

The following buffers were used: equilibration buffer: 50 mM sodium phosphate, 500 mM NaCl, 20 mM imidazole, pH 7.0. Wash buffer: 50 mM sodium phosphate, 500 mM NaCl, 20 mM imidazole, 30% iso-propanol, pH 7.0. Elution buffer: 50 mM sodium phosphate, 500 mM NaCl, 500 mM imidazole, pH 7.0.

Clarified supernatant was loaded to an equilibrated Ni-Sepharose 6 Fast Flow column (GE Healthcare, Uppsala, Sweden) to a capacity of ~30 mg/mL. A residence time of 3 min was used. After loading was completed, the column was washed for 5 column volumes (CV) with equilibration buffer, 5 CV with wash buffer, and 1 CV of equilibration buffer. The bound cpCasp2 was eluted using a linear gradient from 0–100% elution buffer in 10 CV, with a 10 CV hold step to fully elute all protein.

The elution peak of cpCasp2 was buffer exchanged before the polishing chromatography step. Tangential flow ultra-/diafiltration with a 5 kDa cut off membrane was used with a sample buffer of 50 mM sodium citrate, pH 5.0. In total, 5 volumes were exchanged.

The polishing step for wtCasp2 and T7AC-cpCasp2 used cation exchange chromatography on SP Sepharose High Performance. For cpCasp2, Source 30S was used as the stationary phase. The following buffers were used: equilibration buffer A: 50 mM sodium citrate, pH 5.0. Elution buffer B: 50 mM sodium citrate, 1 M NaCl, pH 5.0. Buffer exchanged capture eluate was loaded on the equilibrated polishing column. The residence time was held constant at 1–2 min. The column was loaded to a capacity of ~50 mg/mL. The column was washed for 5 column volumes with 30% B, cpCasp2 was eluted with 10 CV of 45% B and the column was stripped with 3 CV of 100% B. The elution fraction was stored at −80 °C for further use.

### 2.7. Reversed Phase HPLC

Experiments were performed on a Tosoh TSKgel Protein C4-300, L × I.D. 5 cm × 4.6 mm, 3 μm column with a guard column (Tosoh, Tokyo, Japan) on a Waters e2695 HPLC (Waters, Milford, MA, USA). Mobile phase A was water with 0.15% trifluoroacetic acid (TFA) and mobile phase B was acetonitrile with 0.15% TFA. The flowrate was 1 mL/min. Temperature of the column oven was 40 °C, temperature of the autosampler 10 °C. After a 2-min wash, a gradient from 25–50% B in 6 min, followed by a gradient from 50–55% B in 7 min was used to separate cpCasp2 from host cell proteins.

In addition, 200 µL of purified cp caspase-2 (or variant) sample (~4 g/L) was diluted with 100 µL PBS and 100 µL 2 M dithiothreitol (DTT). Furthermore, 10 µl of 0.22 µm filtered sample were injected. The outlet was monitored at 214 nm and 280 nm. The HCP peaks eluted between retention times 3.8 and 9 min. The cpCasp2 peaks eluted between 9.2 and 12.4 min. The peak areas in the 214 nm signal were used to calculate the purity of cpCasp2.

### 2.8. FRET Quantification Assay

A Förster resonance energy transfer (FRET) assay for the determination of the Michaelis–Menten enzymatic activity parameters was performed as described by Öhlknecht et al. [45] with an extended set of substrates. In brief, Michaelis–Menten kinetics were measured by varying substrate concentrations (200, 100, 50, 20, and 10 μM) in 50 mM HEPES, 150 mM NaCl, pH 7.2 at a constant enzyme concentration of 1 μM. The initial slope was determined by measuring the fluorescence for 3 to 15 min (or 3–20 h for proline as P1′, due to the slow kinetics).

The substrates were obtained from Bachem AG (Weil am Rhein, Germany) and were of the general structure of Abz-VDVAD↓XA-Dap(Dnp), where all 20 amino acids were substituted for X (the P1′ position). For specificity testing, substrates with different P1–P4 positions were used. A complete list of substrates can be found in the Appendix A.

### 2.9. Model Substrate Expression and Purification

*E. coli* BL21(DE3) were transformed with the E2 expression plasmid and grown in TY medium (1% peptone, 0.7% yeast extract, 0.25% NaCl) shaking at 37 °C for 16 h. Precultures were diluted in TY medium and further incubated shaking until induction with 1 mM IPTG, at OD_600_ 1.0, executed at 37 °C, 220 rpm, for 4 h. Cell pellets were harvested by centrifugation, and stored at −20 °C until purification.

Frozen cell pellets were resuspended in Tris-buffer (50 mM Tris/HCl, 50 mM NaCl, pH 7.5), disrupted with a French press and the clarified supernatant applied to an IMAC column (HisTrap FF Crude, 1 mL, GE Healthcare, Uppsala, Sweden) connected to an ÄKTA purifier system. Washing was executed for five column volumes with running buffer (50 mM Tris/HCl, pH 7.4, 300 mM NaCl, 20 mM Imidazole). Elution was conducted for five CV with the same buffer containing 250 mM imidazole.

After affinity chromatography imidazole and excess NaCl from pooled peak fractions were exchanged to Tris-buffer with a gel filtration column (HiTrap Desalting, 5 mL, GE Healthcare, Uppsala, Sweden) connected to an ÄKTA purifier system. All elution fractions were pooled, the concentration determined with a BCA assay (Bicinchoninic acid assay, Pierce; BCA Protein Assay Kit, Thermo Fisher Scientific, Waltham, MA, USA), aliquoted, and stored in Tris-buffer with 2 mM DTT at −80 °C.

### 2.10. Cleavage Reaction of Model Substrate

Reactions were prepared with an enzyme to substrate mass ratio of 1:100 (1 mg/mL substrate and 0.01 mg/mL caspase, molar ratio 1:170) in caspase assay buffer (20 mM PIPES, 100 mM NaCl, 10% sucrose, 0.1% CHAPS, 1 mm EDTA, 10 mM DTT, pH 7.2) [39] and incubation at 25 °C. DTT was always added freshly. This setup was defined as standard condition for model substrate cleavage. At defined timepoints, the reaction was stopped and visualized with SDS-PAGE to separate cleaved and uncut protein. Band intensities were analyzed densitometrically to evaluate the time needed for 50% cleavage rate. Under standard conditions, VDVAD-G-E2, which was used to normalize all P1′ substrate cleavages, was cleaved to 50% in 1 min. Substrate was incubated for 24 h without cp caspase-2 to rule out degradation.

### 2.11. N-Terminal Protein Sequencing of the Model Substrate

A sample of a cleavage reaction was subjected to SDS-PAGE (4–12% Bis-Tris Protein Gel, NuPAGE™, Thermo Fisher Scientific, Waltham, MA, USA) to separate caspase, unprocessed substrate, and product and blotted onto a PVDF membrane (poresize, 0.2 µm) by applying 25 V for 1.5 h. XCell II-Blot Module and XCell SureLock-Electrophoresis Cell were used. The blotting buffer contained: 50 mM boric acid, 10% (*v*/*v*) methanol, and was adjusted to pH 9.0 with NaOH (5 M). After complete transfer, the membrane was stained with a Coomassie brilliant blue R-250 solution for several minutes, destained, and washed with water overnight.

The first five N-terminal amino acids of the desired protein band were sequenced by Edman degradation executed by the Protein Micro-Analysis Facility of the Innsbruck Medical University using a Procise Model 492 Edman Micro Sequencer, which was connected online to a Model 140C PTH Amino Acid Analyzer (both from Applied Biosystems, Waltham, MA, USA).

### 2.12. Storage Stability Measurement

T7AC-cpCasp2 was buffer exchanged to different buffers and stored at −20 °C in 20 µL aliquots. After predefined storage time, the samples were thawed and the enzymatic activity was quantified by FRET assay at a constant Abz-VDVAD↓SA-Dap(Dnp) substrate concentration of 100 µM. The following buffers were used: citrate buffered saline (CBS), 50 mM sodium citrate, 450 mM NaCl, pH 5.0 and phosphate buffered saline (PBS), 10 mM Na_2_HPO_4_, 1.8 mM KH_2_PO_4_, 137 mM NaCl, 2.7 mM KCl, pH 7.4. Both buffers were prepared with and without the addition of 0.05% (*v*/*v*) polysorbate 20 (TWEEN20).

For the freeze thaw/stability study, T7AC-cpCasp2 in CBS buffer was stored at −20 °C until completely frozen. The sample was subsequently thawed and enzymatic activity was quantified by FRET assay at a constant Abz-VDVAD↓SA-Dap(Dnp) substrate concentration of 100 µM.

## 3. Results and Discussion

### 3.1. Circular Permutation of Caspase-2

A circularly permuted human caspase-2 (cpCasp2) was designed in order to obtain a constitutively active single chain protease that can easily be expressed in *E. coli*. After the activating cleavage between the subunits of wild type (wt) Casp2 (Figure 1a), the C-terminus of the large and the N-terminus of the small subunit move apart, while the C-terminus of the small and the N-terminus of the large subunit stay close together [46]. This proximity has been utilized to engineer a cpCasp2. The order of large and small subunits in the primary structure of the circularly permuted caspases is switched and the small subunit now precedes the large (Figure 1b). This facilitates folding into the active conformation without cleavage. To the cpCasp2 with the two linked sub-units, a His-tag was added N-terminally.

Circular permutation of caspase-2 improved the manufacturability in *E. coli* leading to a drastic increase in recombinant protein expression. A direct comparison of wtCasp2 production with cpCasp2 production regarding cell growth and soluble and insoluble recombinant protein production in lab-scale bioreactors highlighted that overexpression of cpCasp2 was possible in *E. coli*, whereas the expression of soluble wtCasp2 was generally low and only estimable with Western blot analysis (Figure 2). Formation of wtCasp2 IB was not observed. In principle, caspases are active in *E. coli* and can potentially cleave bacterial proteins and negatively influence growth and yield [21]. In our case, the product titers did not cause a reduction of growth rate. As caspases have a relatively complex 3D structure, we assume a complex in vivo folding pathway until an active protease is produced. Cell growth of both wtCasp2 and cpCasp2 processes followed the calculated CDM. Final CDM was about 70 g/L. In total, 8.2 g/L cpCasp2 were produced in a HCD fermentation, but the majority was present in IBs and only 430 mg/L were obtained in soluble form. This corresponds to a specific protein content of 6.3 mg/g. In comparison, wtCasp2 was expressed with less than 1 mg/g.

**Figure 1 biomolecules-10-01592-f001:**
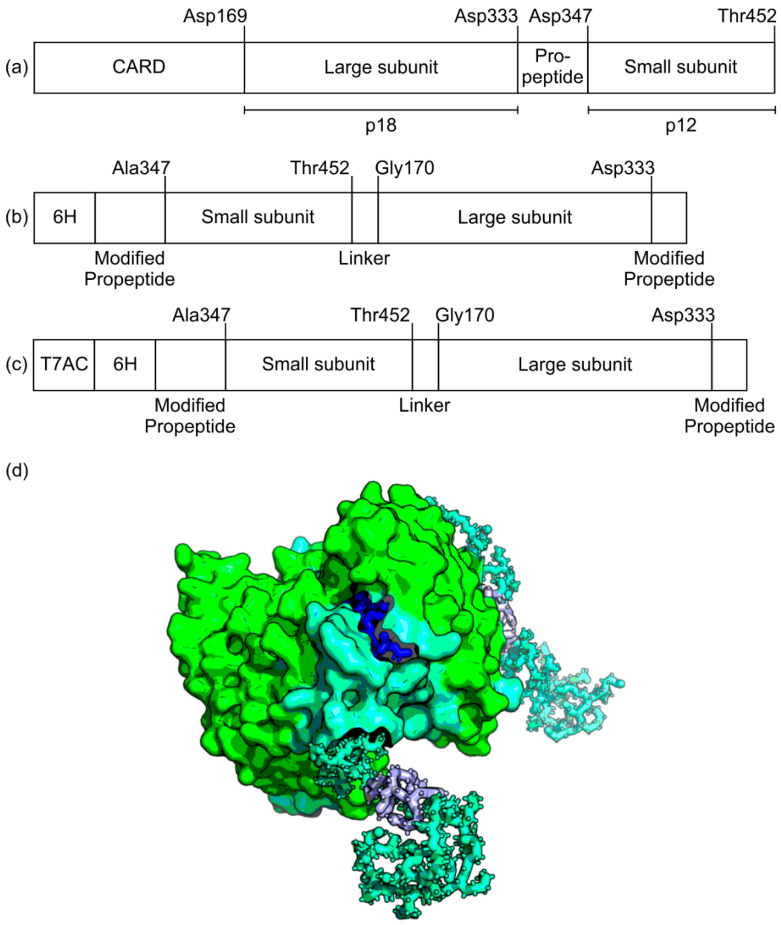
(**a**) schematic representation of wtCasp2. Aspartate residues critical for activation are annotated. Length of subunits is not true to scale. By cleavage after Asp333, the fully active enzyme is formed, and the mature caspase is obtained by two further cleavage events at Asp169 and Asp347 [47]. Cleavage at Asp347 reduces the small subunit from a p13 to a p12; this, however, does not further increase activity [18]. (**b**) In the cp variant, the order of subunits is reversed. The pro-peptide before the small subunit is modified to avoid autocatalytic cleavage. (**c**) For T7AC-cpCasp2, the T7AC tag is added N-terminally; (**d**) crystal structure of a T7AC-cpCasp2 dimer, based on PDB structure 1PYO with added N-terminal tags [14]. The large subunit is green, while the small subunit is turquoise. The his-tag is shown in purple and a substrate peptide is shown bound to the active site in blue.

**Figure 2 biomolecules-10-01592-f002:**
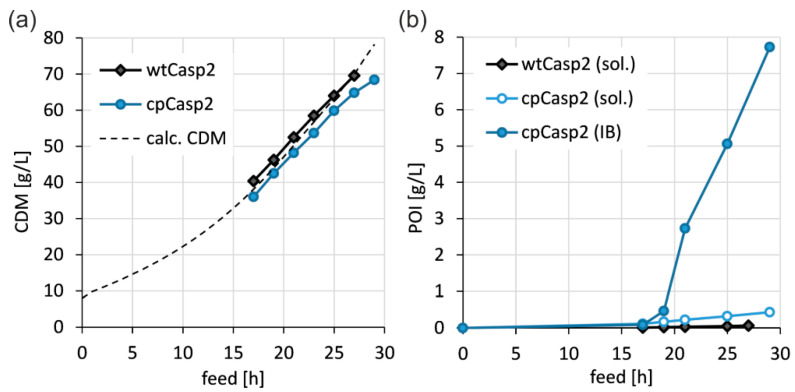
Direct comparison between wtCasp2 and cpCasp2 production during carbon limited fed-batch cultivation (µ = 0.1 h^−1^) with constant 0.5 µmol IPTG/g CDM: biomass course: cell growth (**a**) and expression; (**b**) of soluble (sol.) and insoluble (IB) POI in the course of time as volumetric titer [g/L].

A simple two step downstream process was set up, with an IMAC capture step after cell lysis by high pressure homogenization, followed by a CIEX polishing step. The purity of cpCasp2 was 98.8%, as determined by RP-HPLC. In total, 0.05 mg cpCasp2 could be purified from 1 g of CDM, which is a 3.5-fold increase from the 0.014 mg wtCasp2 that could be purified from the same amount of CDM. Nevertheless, the overall yield of cpCasp2 was low because of high IB formation. The losses during DSP were assumed to be high but could not be quantified owing to the lack of a sensitive quantitative method for cpCasp2.

Both enzymes were active, as determined by a peptide-based FRET assay, with cpCasp2 exhibiting higher activity, which can be seen in Section 3.3. Characterization of Caspase-2.

### 3.2. Production of T7AC-cpCasp2

A solubility tag, a peptide extension with a large net negative charge designed analogous to the bacteriophage T7, was fused N-terminally to cpCasp2 to increase soluble protein yields (Figure 1c). The tag was based on the T7A3 peptide that can increase electrostatic repulsion between nascent polypeptides and thereby limits protein aggregation as described by Zhang et al. [33]. We modified the tag to remove potential Casp2 cleavage sequences. In brief, after in silico analysis, two Glu residues were mutated to Gln, to avoid autocatalytic cleavage of the tag by cpCasp2. Even though this reduces the net charge of the new T7AC tag to −3 (not counting the C-terminus), it maintains its highly hydrophilic nature. In HCD fermentations, a positive influence of the solubility tag could be shown, and expression of T7AC-cpCasp2 was slightly shifted away from IB formation (Figure 3). Total expression stayed at around 8 g/L, with soluble protein titer of 1 g/L. A specific protein titer of 13 mg/g was reached. Strikingly, N-terminal fusion of the T7AC solubility tag to cpCasp2 led to a two-fold increase in soluble titer. Moreover, IMAC purification and enzymatic activity were not negatively influenced by addition of the T7AC tag and the enzyme could be captured by metal chelate chromatography and used for fusion protein processing.

High recombinant expression of a POI leads to stressful situations for the host cell, resulting in protein misfolding in vivo, and consequent aggregation into IBs [48]. Several methods have been shown to prevent protein aggregation as reviewed by Costa et al. [25]. In our study, we attempted to develop fermentation and induction strategies for successful and efficient soluble protein production. A series of cultivation runs were conducted according to a Design of experiments (DoEs) approach (Figure 4a) with the production clone BL21(DE3) (pET30a*cer*-T7AC-cpCasp2). The influence of different growth rates (µ = 0.03, 0.05 and 0.07 h^−1^) and induction strengths (0.5, 0.9 and 1.3 µmol IPTG/g CDM) were investigated regarding cell growth (Figure 4b) and soluble and insoluble recombinant protein production (Figure 4c,d).

A decrease of the growth rate caused higher specific soluble T7AC-cpCasp2 titers up to 100 mg/g CDM (Figure 4c) leading to lower biomass concentrations over the whole fermentation (Figure 4b). The observed cell mass at the lowest growth rate was only around 60% of the calculated cell mass. With the highest growth rate, cell growth followed the calculated CDM (Figure 4b). The final CDM was about 77.5 g/L, respectively, 1549 g in total. The highest volumetric soluble titers of about 5 g/L were observed for the lowest growth rate (0.03 h^−1^) and low inducer concentrations (0.5 and 0.9 µmol IPTG/g CDM) (Figure 4d). By lowering the growth rate and the strength of induction, the physiological stress could be reduced at the cost of increased process time.

For final process optimization, the fermentation strategy was further adjusted. The medium was calculated for a higher cell dry mass, the production phase was shortened, and a low growth rate of 0.03 h^−1^ and a medium inducer concentration of 0.9 µmol IPTG/g CDM were chosen. Under these conditions, total recombinant production increased to 10.7 g/L, with 50% soluble protein fraction leading to 5.3 g/L soluble T7AC-cpCasp2. Figure 5 summarizes the development of the caspase-2 production process.

The higher soluble protein yield of 5.3 g/L increased specific soluble protein content to 78 mg/g and enzyme quantification by FRET assay allowed the optimization of DSP. The high-pressure homogenization was evaluated, and an optimal cell lysis could be achieved at 1400 bar and two passages. Losses during both chromatography steps due to irreversible adsorption were observed. The stationary phase for the CIEX polishing step was changed from SOURCE to Sepharose media, which decreased irreversible adsorption losses on the more hydrophilic backbone [49]. An overall process yield of 38% could be achieved with a purity of ~99% according to RP-HPLC, with the highest losses during IMAC capture (Table 1 and Appendix A). This results in a high manufacturability of 29 mg purified T7AC-cpCasp2 per g CDM, or a volumetric yield of 2 g purified enzyme per L fermentation. This is a more than a 500-fold increase from the manufacturability of cpCasp2, where only 0.05 mg purified cpCasp2 per g CDM were produced.

### 3.3. Characterization of Caspase-2

Caspase-2 cleaves after the amino acid sequence VDVAD, even if followed by proline [17,50], but cleavage kinetics are influenced by the amino acid in the P1′ site. This influence of the P1′ amino acid on the cleavage kinetics was investigated by measuring Michaelis–Menten kinetics with a peptide substrate-based FRET assay. The substrates had the general structure of fluorophore-VDVAD↓XA-quencher, with X being any amino acid (Figure 1c). For wtCasp2 only G, F, Q, and V could be determined, due to the low sample amount. Figure 6a shows k_cat_/K_M_ for all three Casp2 variants. The Michaelis–Menten kinetics are highly dependent on the P1′ amino acid, with glycine exhibiting the highest k_cat_/K_M_ and proline the lowest. The k_cat_/K_M_ values span almost five orders of magnitude for both cpCasp2 and T7AC-cpCasp2, as shown in Table 2. Full values can be found in the Appendix A. The differences in k_cat_/K_M_ are mostly driven by k_cat_ and not by K_M_, which indicates that the P1′ site strongly influences the rate of the cleavage reaction but not the affinity (see Supplementary Material File S6).

T7AC-cpCasp2 has the highest k_cat_/K_M_ values for all substrates and wtCasp2 the lowest for those that were determined. This increase in activity is largely driven by k_cat_, while K_M_ values are similar. On average, k_cat_ values of cpCasp2 are four times higher than of wtCasp2, and T7AC-cpCasp2 exhibits k_cat_ values twice as high as cpCasp2. We assume that the aggregation propensity of T7AC-cpCasp2 is lower compared to cpCasp2 which might explain the difference in k_cat_ values. Alternatively, due to the differences in upstream and downstream processing yield, it is possible that these differences are not of a structural nature but are due to other factors such as co-purified compounds. The observed catalytic efficiency (k_cat_/K_M_) for the substrate VDVAD↓GA is higher than that of other specific proteases, such as SUMO (782 M^−1^·s^−1^) and TEV (28 M^−1^·s^−1^) [51], but lower than less specific proteases such as thrombin (8600 M^−1^·s^−1^) [52] or enteropeptidase (719,000 M^−1^·s^−1^) [53].

A model protein cleavage was chosen in addition to the peptide-based FRET measurements to simulate realistic process conditions and to evaluate the specificity and target activity of cpCasp2. Ubiquitin-conjugating enzyme E2 L3 (E2) was chosen and expressed with the N-terminal tag 6H-GSG-VDVAD↓X, with X being all 20 amino acids. Figure 6b shows a plot of E2 cleavage with cpCasp2. The time to cleave 50% of each substrate was measured. The time required for cleavage ranges from several minutes to several hours depending on the P1′ residue. All values can be found in the Appendix A. This time is compatible and economic for laboratory and industrial processes [54]. The order of cleavage efficiency is very similar for the peptide and the protein substrates. When comparing Figure 6a,b, some P1′ amino acids exhibit differences from peptide to protein cleavage assay, most notably, F, S, and I. We hypothesize that cleavage kinetics are not only determined by P1′ amino acid, but also by the overall structure of the cleavage region, i.e., the N-terminus of the native protein of interest. Edman sequencing of processed E2 showed that an authentic N-terminus was generated after tag removal and no off-target cleavage was observed. Processed E2 was also analyzed by SDS-PAGE, and no additional bands indicated unspecific cleavage events. To our knowledge, this is the first characterization of P1′ dependence of the cleavage reaction in caspase-2 with peptide and protein substrates.

The main requirements for processing of fusion proteins at industrial scale are specificity and creation of an authentic N-terminus. Most available enzymes do not even fulfill these basic demands. The frequently used proteases thrombin and factor Xa, for example, are both known to cleave proteins at non-specific sites [1,9,10].

The protease presented here exhibits very high specificity as shown in Figure 7, where substrates with sub-optimal cleavage sites showed cleavage efficiencies lower than 1% compared to the canonical VDVAD sequence. This means that potential side reactions, resulting in unspecific cleavage are many times slower than the tag removal reaction, leading to no loss of product.

Several alternative cleavage sites were chosen to assess the influence of diverse residues in the recognition sequence on cpCasp2. The influence of the P5 residue was tested with cleavage site DEVD, preferred by caspase-3 and -7, and confirmed the higher specificity over tetrapeptide recognition sites. Recognition sites EISD, EFKD, and HYID were chosen to evaluate the influence of the P4 residue where usually an aspartate is preferred by caspases. The drastically reduced cleavage times for both sites again demonstrate the reduced probability for off-target cleavages. VDQQE and VDQQS were designed to test the possibility of C-terminal addition of tags to cpCasp2. VDQQD are the last amino acids in cpCasp2 and constitute the C-terminus of the large subunit, the site of activation by autocatalytic cleavage in wtCasp2. The P1 aspartate was mutated to glutamate and serine in an attempt to prevent possible cleavage. The VDQQS mutation is hardly recognized, which could enable the addition of a tag at this terminus. The large subunit’s C-terminus is annotated as a pro-peptide in the original caspase-2 structure (UniProt ID P42575). Residues 326–333 in the wt and equal amino acids 286–292 in cpCasp2 suggest a potential cleavage site DETD↓R between large subunit p18 and propeptide/modified propeptide (see Appendix A). Possible cleavage was investigated by mutation of the P1 aspartate to glutamate.

Storage stability of T7AC-cpCasp2 was investigated in different buffer systems at −20 °C for two years. The stability was evaluated by enzymatic activity and a sample stored at −80 °C was used as control. Figure 8 shows that T7AC-cpCasp2 exhibits excellent long-term storage stability in citrate or phosphate based saline buffer systems with 0.05% polysorbate (TWEEN) 20. Alternatively, similar stability can be reached without the addition of surfactant at −80 °C. The resistance to freeze/thaw degradation can be seen in Figure 8b, were no significant decrease in activity could be detected over 25 freeze/thaw cycles (Student’s *t*-test, two-tailed *p*-value = 0.1123, *n* = 9, α = 0.95).

## 4. Conclusions

We generated an enzyme for processing of fusion proteins which fulfills all essential elements for potential industrial applications, namely high expression yield, simple production, high selectivity, fast kinetics, and high stability. We obtained such an enzyme by circularly permuting caspase-2 and equipping the enzyme with a solubility and an affinity tag and developed a laboratory scale production process. We are able to overexpress the protein to a high yield with 5 g active enzyme per liter. T7AC-cpCasp2 represents a protease that has a high specificity similar to TEV protease, but with higher activity and high manufacturability, with high freedom for the N-terminal amino acid in the P1′ site. As such, it allows for the generation of a native N-terminus similar to Factor Xa, Enterokinase, subtilisin or SUMO protease, without off-target cleavage or requiring a very large tag. The extremely high stability without loss of enzyme activity at −20 °C for one year, a high specificity, and cleavage activity irrespective of the amino acid at the P1′ site make the enzyme suitable for a platform process.

## Figures and Tables

**Figure 3 biomolecules-10-01592-f003:**
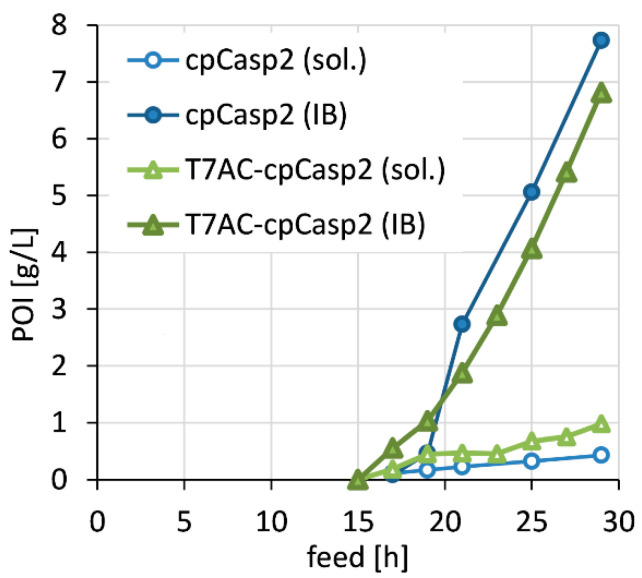
Direct comparison between cpCasp2 production with (T7AC-6H-cpCasp2) and without (6H-cpCasp2) solubility tag T7AC during carbon limited fed-batch cultivation (µ = 0.1 h^−1^) with constant 0.5 µmol IPTG /g CDM: expression of soluble (sol.) and insoluble (IB) POI in the course of time as volumetric titer [g/L].

**Figure 4 biomolecules-10-01592-f004:**
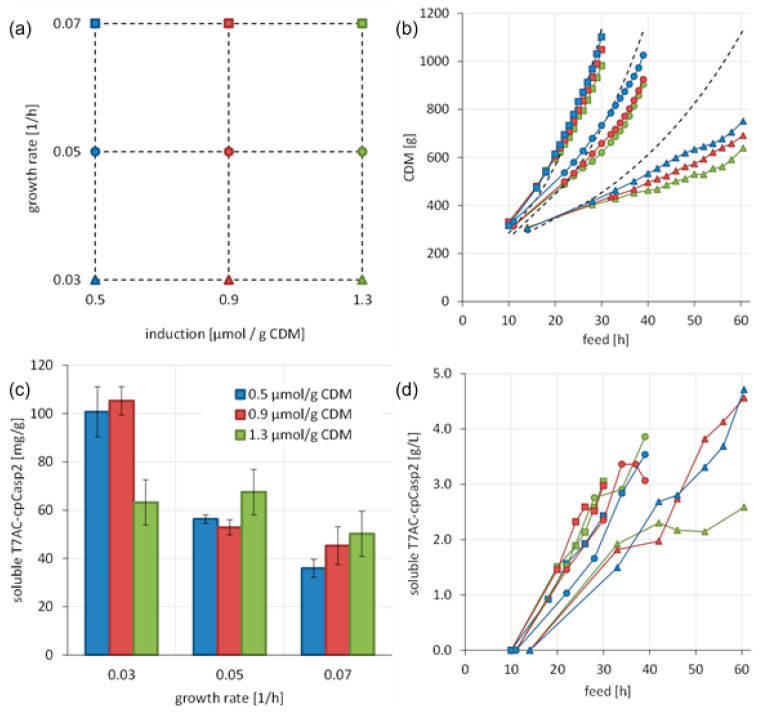
DoE approach to improve soluble T7AC-cpCasp2 expression: nine cultivations of *E. coli* BL21(DE3)(pET30a-T7AC-cpCasp2) during carbon limited 2 phase fed-batch cultivation, (growth rate µ = 0.17, followed by different growth rates during induction, µ = 0.03, 0.05 and 0.07 h^−1^) with three different IPTG induction strengths (0.5, 0.9 and 1.3 µmol IPTG/g CDM). (**a**) Overview of parameters that were varied with legend to colors and symbols used in the other panels. The colors correspond to the induction strength and the symbols correspond to the growth rate. (**b**) Cell growth at different conditions. (**c**) Specific titer at the end of the cultivation; mean values and experimental standard deviation are shown (*n* = 3). (**d**) Soluble expression in the course of time as volumetric titer [g/L].

**Figure 5 biomolecules-10-01592-f005:**
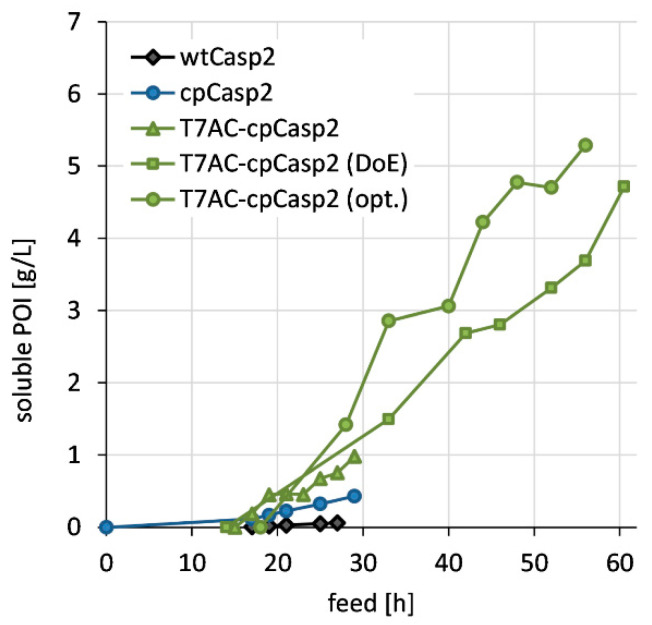
Summary of all engineering and fermentation development steps for the efficient production of soluble of cp caspase-2.

**Figure 6 biomolecules-10-01592-f006:**
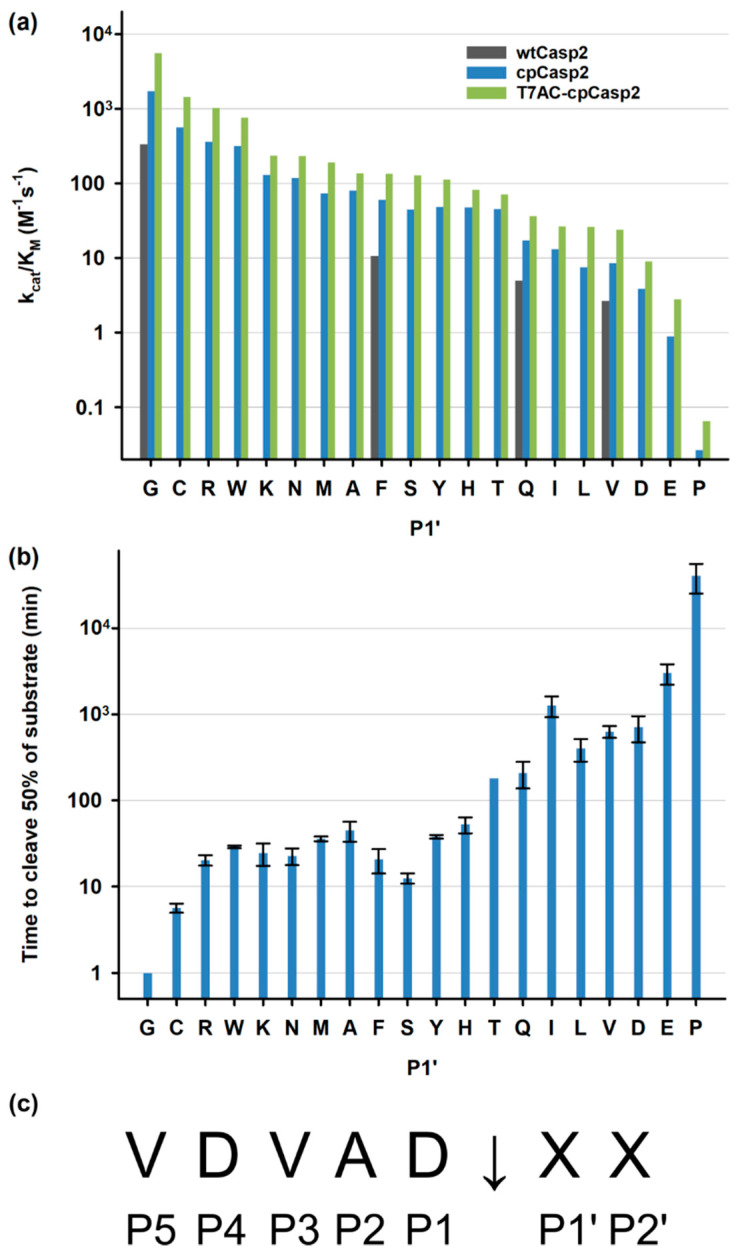
(**a**) FRET enzyme kinetics of wtCasp2, cpCasp2 and T7AC-cpCasp2. Due to the low available amount of wtCasp2 only G, F, Q and V were measured; (**b**) cleavage of the protein substrate E2 with all 20 amino acids in the P1′ position, using cpCasp2. Error bars are experimental standard deviation (n = 3); (**c**) the recognitions sequence of Casp2 denoting the naming convention used in this work. ↓ marks the cleavage site.

**Figure 7 biomolecules-10-01592-f007:**
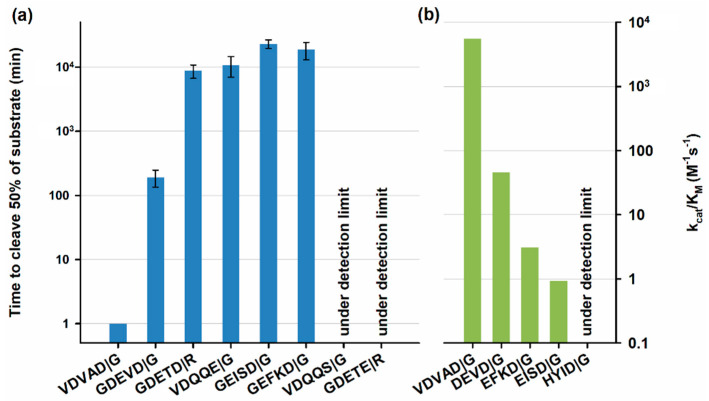
(**a**) specificity of cpCasp2 shown by cleavage of E2 substrate with varying recognition sites. Error bars are experimental standard deviation (*n* = 3); (**b**) specificity of T7AC-cpCasp2 shown by cleavage of peptide substrates.

**Figure 8 biomolecules-10-01592-f008:**
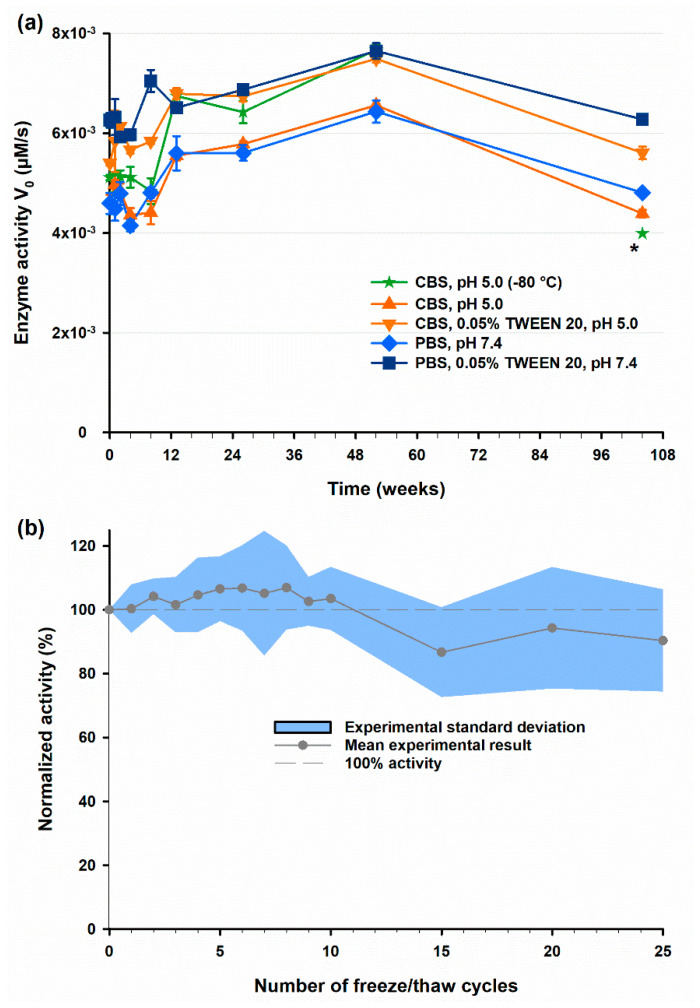
(**a**) long-term storage stability of T7AC-cpCasp2, as determined by FRET assay. Different formulations were stored at –20 °C and compared to a control sample stored at –80 °C. Error bars are experimental standard deviation (*n* = 3). *For the 104-week data point of CBS, pH 5.0 (–80 °C) a different production lot of T7AC-cpCasp2 with a slightly lower base activity had to be used; (**b**) enzymatic activity over 25 freeze/thaw cycles. T7AC-cpCasp2 was repeatedly frozen at –20 °C and thawed. Enzymatic activity was determined by FRET assay. The blue area denotes the standard deviation of the experiment (*n* = 9). No significant activity decrease could be detected after 25 freeze/thaw cycles.

**Table 1 biomolecules-10-01592-t001:** Step yields of the T7AC-cpCasp2 DSP.

Process Step	Step Yield	Purity (HPLC)
primary recovery	97%	n.d.
IMAC capture	63%	96.5%
UF/DF buffer exchange	84%	n.d.
CIEX polishing	74%	99.2%

n.d. = not determined.

**Table 2 biomolecules-10-01592-t002:** Selected Michaelis–Menten parameters of peptide cleavage with Casp2 variants.

		P1′				
		F	G	P	Q	V
wtCasp2	K_M_ [µM]	79 ± 11	97 ± 12	n.d.	114 ± 10	86 ± 9
	k_cat_ [s^−1^]	8.4 ± 0.5 × 10^−4^	3.2 ± 0.2 × 10^−2^	n.d.	5.7 ± 0.2 × 10^−4^	2.3 ± 0.1 × 10^−4^
	k_cat_/K_M_ [M^−1^s^−1^]	11	335	n.d.	5.0	2.7
cpCasp2	K_M_ [µM]	60 ± 16	112 ± 37	305 ± 146	121 ± 24	64 ± 23
	k_cat_ [s^−1^]	3.6 ± 0.4 × 10^−3^	1.9 ± 0.3 × 10^−1^	8.1 ± 2.6 × 10^−6^	2.1 ± 0.2 × 10^−3^	5.4 ± 0.8 × 10^−4^
	k_cat_/K_M_ [M^−1^s^−1^]	61	1720	0.026	17	8.5
T7AC-cpCasp2	K_M_ [µM]	58 ± 15	49 ± 13	152 ± 66	128 ± 24	73 ± 18
	k_cat_ [s^−1^]	7.9 ± 0.8 × 10^−3^	2.7 ± 0.3 × 10^−1^	9.9 ± 2.4 × 10^−6^	4.6 ± 0.5 × 10^−3^	1.7 ± 0.2 × 10^−3^
	k_cat_/K_M_ [M^−1^s^−1^]	136	5542	0.065	36	24

Values for K_M_ and k_cat_ are given as measured value plus 95% confidence interval (*n* = 15). Kinetics for proline P1′ substrate with wtCasp2 could not be determined due to lack of purified enzyme. n.d. = not determined.

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
