# Peer review of "Production of Circularly Permuted Caspase-2 for Affinity Fusion-Tag Removal: Cloning, Expression in Escherichia coli, Purification, and Characterization"

_biomolecules, 2020, doi:10.3390/biom10121592_

Round 1

Reviewer 1 Report

Here are more specific questions for the authors:

  1. In general, majority figures lack statistic information. In figures 2, 3, 4b, 5, 6a, 7b and 8b, the number of repeated experiments and error bars should be shown. Without repeats and error bars, some data is not convincing, especially for those cases that difference is within an order of magnitude.
  2. Citations should be added to demonstrate unspecific cleavage of other proteases in line 63 and 64.
  3. In figure 1, a crystal structure should be shown to clearly indicate domains in caspase-2 if the structure is available.
  4. In figure 1, the unique cleave site with clear labeled “P1” position should be included. After reading through the paper, I understand the “P1” position is quite important for the enzyme activity. I will appreciate more if this important information is shown much earlier.
  5. T7AC peptide seems play an important role to solubilize the cpCASP2 when over expressed in E. coli. Did you test if the T7AC peptide can improve expression and solubility of the wtCASP2? I’m wondering if the overall improvement of the caspase-2 is due to circular permutation or fusion of soluble peptide like T7AC.
  6. In Row 372, the author mentioned the reference to Figure 6. However, figure 6 is an information dense figure. Be more specifically with your reference will be helpful.

Reviewer 2 Report

The authors provide an excellent new caspase enzyme for the proteolytic removal of affinity tags in the purification of recombinant proteins.  The study includes a optimization of the expression and purification of the protein, as well as a characterization of the caspase with two model substrates.  The experimental descriptions include many references to previous work, which makes them a little difficult to follow, but at first reading it appears that sufficient detail has been provided to reproduce the work.  The authors are correct that no universally useful protease enzyme exists for tag removal, and this caspase may therefore become an important tool for this application.  The overall conclusions of the work would be made more compelling with some additional information though.

A significant concern is that the caspase activity has only been characterized for a small unstructured peptide (for the FRET P1’ experiments), and a single model target protein.  It is also notable that the small substrate kinetics do not completely match the kinetics of the model protein for all P1’ residues, suggesting that the P1’ residue may not be the only determinant for cleavage rate.  This should be mentioned and discussed, especially for the residues where the difference in cleaving rate is substantial (as it is for I and L for example).  It would greatly increase the faith in this caspase to be reliable and general if consistent activity could be shown for additional model proteins.

The caspase assay buffer has PIPES, sucrose, CHAPS and DTT (as well as EDTA and NaCl).  Which of these components are required for activity?  How was this buffer selected?  In particular, the potential requirement for DTT is concerning due to its potential effects on disulfide bonds in the POI, while CHAPS and sucrose are also unusual for a bioprocess at large scale.  This should be discussed.  The buffer conditions for the FRET assay are not provided and should be added as well.

Additional suggested corrections:

Some cosmetic errors should be corrected, like E. coli should be italicized in lines 45, 106 and 107 and in other locations.

It would be helpful to have a figure showing the cleaving target sequence and defining the P1 and P1’ positions.  These are not clear in the text, and not all readers will be familiar with this naming convention.  This could be added as a panel to Figure 6.

Figure 1 should be referenced in the Materials section when the cyclization strategy is described.  Figure 1 should also be modified to show the position of the solubility tag, and its sequence if possible.  I would also propose to put the full T7AC-cpCasp2 sequence in the main manuscript with the different features labeled, or at least have the different structural features more clearly marked in the Supplemental materials sequences.

CDM should be defined the first time it is used.  It should also be explained how the CDM was determined experimentally (calculation from growth medium, extrapolation from OD, or direct measurement?).

It seems that the optimized fermentation provided 108 g CDM/L fermentation, and the cells were lysed at a concentration of 30 g CDM/L lysis buffer.  It would appear that the lysis volume would have been about 30 L then, which seems like a lot for a JLA-10.500 rotor.  Can the authors clarify the actual purification scale that was used in their experiments?

It would be nice to see some actual HPLC data or a gel to show the purity of the caspase over the course of the purification.  This could be put in the Supplemental section if desired.

Line 515 has a reference error in it.

Reviewer 3 Report

This manuscript describes the expression, purification and characterization of a new circularly permuted human caspase-2 (cpCasp2) enzyme to be used for affinity tag removal in a platform fusion protein process. The protease has been purified with an extremely high yield, and its stability and specificity has been demonstrated. It is only prone to carry out non-specific cleavages after a prolonged period of incubation, which is, however, needed for certain amino acids at P1' site. This protease can be of interest for the protein research society, and the manuscript is worth publishing. 

My comments: 

-Several reasons for searching a new protease for affinity tag cleavage were listed in the introduction. In my opinion, the strategies to avoid aberrant or non-natural N-terminus of the protein of interest  shall be collected from the literature for comparison. 

-A scheme of the protease cleavage labelling the P1 and P1' sites would help understanding. 

-It is not clear, how the non-specific cleavage sequences shown in Fig. 7 were selected. 

-In lane 515 there is and editing error: Error! Reference source not found

-The properties of the protein of interest may influence the applied strategy for its purification. It may not be soluble/stable under the conditions of the protease cleavage. Is there any information about the behaviour of the new protease in various buffers? 

-A more detailed discussion about the advantages and disadvantages of the new protease in the conclusion part of the manuscript would help the potential users of the new platform to establish a successful experimental protocol. 

Round 2

Reviewer 1 Report

I think the authors nicely addressed my comments. I don't have further comments on this manuscript.

Reviewer 2 Report

The additional purification data is very helpful, although it raises some questions about the homogeneity of the caspase product.  Answering these questions is probably beyond the scope of the manuscript though, so I will accept that the authors have addressed my concerns.